# A Rare Case of Solitary Neurofibroma Misdiagnosed as Diabetic Foot Ulcer in the Toe Tip Region

**DOI:** 10.3390/medicina60081200

**Published:** 2024-07-24

**Authors:** Ha-Jong Nam, Se-Young Kim, Mee-Hye Oh, Soomin Lim, Hwan-Jun Choi

**Affiliations:** 1Department of Plastic and Reconstructive Surgery, Soonchunhyang University Gumi Hospital, Gumi 39371, Republic of Korea; 125039@schmc.ac.kr (H.-J.N.); 111459@schmc.ac.kr (S.-Y.K.); 2Department of Pathology, Soonchunhyang University Cheonan Hospital, Cheonan 31538, Republic of Korea; mhoh0212@schmc.ac.kr; 3Medical School, University College London (UCL), London WC1E 6DE, UK; iamsoominlim@gmail.com; 4Department of Plastic and Reconstructive Surgery, Soonchunhyang University Cheonan Hospital, Cheonan 31538, Republic of Korea

**Keywords:** neurofibroma, foot ulcer, diabetic, diabetes mellitus, type 2, case reports

## Abstract

Neurofibromas, rare benign tumors of the peripheral nerve sheath, present diagnostic challenges, particularly in diabetic patients with toe ulcers. This case involves a 55-year-old female with type 2 diabetes mellitus who developed an enlarging ulcer on her right second toe. The initial evaluation suggested a diabetic ulcer; however, advanced imaging revealed a mass-like lesion. Partial excision and biopsy confirmed a neurofibroma with spindle cells within the myxoid stroma and S100 protein expression. One month later, total excision and Z-plasty reconstruction were performed under general anesthesia. The patient’s postoperative recovery was uneventful, and the patient was discharged without complications. Follow-up revealed successful healing with no recurrence or functional issues. This case highlights the importance of considering neurofibromas in the differential diagnosis of diabetic toe ulcers to avoid misdiagnosis and ensure appropriate management.

## 1. Introduction

Neurofibromas are benign tumors that originate from the peripheral nerve sheath and constitute approximately 5% of all benign soft tissue tumors [1]. Solitary neurofibromas, which occur without an association with neurofibromatosis, are particularly rare and typically present as painless slow-growing masses [2]. These tumors pose significant diagnostic challenges, particularly in patients with coexisting conditions.

Diabetic foot ulcers, which frequently result from peripheral neuropathy and vascular insufficiency, are a common and serious complication of diabetes mellitus [3]. These ulcers usually present with infection, induration, and delayed healing, often necessitating a multidisciplinary approach for effective management [4]. However, the assumption that all toe tip ulcers in patients with diabetes are related to diabetic foot pathology can lead to misdiagnosis and inappropriate management strategies.

This case report highlights a rare instance of a solitary neurofibroma presenting as a toe tip ulcer in a patient with diabetes initially misdiagnosed with a diabetic foot ulcer. Despite the initial misdiagnosis, the diabetic foot evaluations, including peripheral neuropathy and vascular assessments, are relatively safe and cost-effective, justifying their use as first-line diagnostics [5]. However, the presence of atypical features necessitates further investigation to exclude other tumors. This case underscores the importance of considering a wide range of differential diagnoses, even in the presence of common diabetic complications. By detailing the clinical presentation, diagnostic workup, and successful management of this patient, we aimed to enhance awareness of this rare entity and its potential for misdiagnosis.

## 2. Case Presentation

A 55-year-old woman with a history of type 2 diabetes mellitus, diagnosed 5 months prior and managed with metformin and dietary education, presented with a gradually enlarging ulcer on the right second toe tip over the past 2 months. Physical examination revealed substantial enlargement, induration, and discharge of the toe tips (Figure 1A). The toe was swollen and showed signs of inflammation; however, there were no signs of systemic infection. Diabetic foot evaluation, including the assessment of peripheral neuropathy and vascular insufficiency, revealed that all test results were within the normal range.

Initial imaging studies included an X-ray, which showed no specific findings in the bony structure (Figure 1B). However, lower extremity angiography-enhanced computed tomography (CT) revealed a mass-like lesion with feeding vessels (Figure 2A). Magnetic resonance imaging (MRI) depicted a cystic lesion, raising the suspicion of a soft tissue tumor rather than a simple diabetic ulcer (Figure 2B). Laboratory tests showed that blood parameters were within normal limits, except for elevated levels of HbA1c and fasting blood glucose, as well as slightly increased levels of inflammatory markers. Detailed laboratory results, including Hemoglobin A1c (HbA1c) and glycemia upon admission, are presented in Table 1.

The patient initially underwent a partial excision and biopsy of the lesion. Histopathological examination confirmed the presence of neurofibroma, characterized by interlacing bundles of spindle cells with wavy, darkly stained nuclei, myxoid stroma and collagen fibers. The tumor showed strong positivity in the nucleus and cytoplasm for S100 protein (Figure 3A,B). The patient continued her metformin regimen and dietary education throughout the perioperative period, except during the period of contrast agent use for CT imaging. There were no significant fluctuations in blood glucose levels during her hospital stay, and her HbA1c remained stable postoperatively. One month later, the patient was readmitted for the total excision of the mass under general anesthesia. During the surgery, a neurovascular bundle was observed, meticulously skeletonized, and separated before the mass was evacuated (Figure 4A). The excised mass showed a sheath encasing the neurofibroma (Figure 4B). Reconstruction of the toe tip shape was achieved using Z-plasty (Figure 4C).

The patient was discharged without wound-related complications one week postoperatively. Follow-ups revealed no scar contractures, functional issues, or sensory abnormalities. The patient reported no symptom recurrence or ulceration at subsequent visits (Figure 5A). Two months postoperatively, the toe appeared to be well healed without any signs of infection or other adverse effects (Figure 5B). This case illustrates the importance of considering neurofibromas in the differential diagnosis of toe tip ulcers, particularly in patients with diabetes, to avoid misdiagnosis and ensure appropriate management.

## 3. Discussion

Solitary neurofibromas are rare benign tumors originating from the peripheral nerve sheath, with even rarer occurrences in the toe tip region [6]. These tumors typically present as painless, slow-growing masses, and do not commonly cause ulceration or induration [2]. Due to their asymptomatic nature and benign appearance, it is challenging to immediately recognize them as neurofibromas when they occur in uncommon locations such as the toe tip. This diagnostic challenge is heightened in patients with diabetes in whom foot lesions are more commonly attributed to diabetic complications [7].

In this case, the patient’s initial diagnosis of diabetic foot ulcer was reconsidered following advanced imaging studies, which revealed a mass-like lesion with feeding vessels. This finding, combined with the MRI results indicating a cystic lesion, prompted a differential diagnosis that included soft tissue tumors. The initial partial excision and biopsy confirmed the presence of a neurofibroma, leading to subsequent total excision and toe reconstruction.

Despite initial misdiagnosis, diabetic foot evaluation, including peripheral neuropathy and vascular assessment, is relatively safe and cost-effective. These tests are justifiable as first-line diagnostic tests because of their efficiency and low risk [8]. However, the presence of atypical features in this case highlights the necessity of ruling out other tumors.

The potential for misdiagnosis in patients with diabetes is significant, given that foot ulcers are a common complication of diabetes [9]. This case underscores the importance of comprehensive diagnostic evaluation, including imaging and biopsy, when typical diabetic foot ulcer treatments fail or when unusual characteristics are present. Early identification and appropriate surgical intervention are crucial to prevent complications and ensure favorable outcomes [10].

The successful management of this patient underscores the importance of a multidisciplinary approach involving radiologists, pathologists, and surgeons to accurately diagnose and treat solitary neurofibromas. In the present case, a lesion initially suspected to be a routine diabetic foot infection was promptly identified through early histological examination, allowing for a precise pathological diagnosis. This facilitates timely and appropriate surgical excision, resulting in an aesthetically superior toe reconstruction. This case highlights the importance of early histopathological assessment and targeted surgical intervention in achieving optimal functional and cosmetic outcomes in reconstructive surgery. Thorough investigation of lesions, including those initially suspected to be straightforward infections, is essential to ensure accurate diagnosis and effective treatment.

One limitation of this case report is the inability to generalize the findings owing to the singular nature of the case. Additionally, the lack of long-term follow-up data limits our understanding of potential recurrence or late-onset complications. However, despite the absence of long-term follow-ups, the patient demonstrated an aesthetically pleasing outcome with no adverse effects or complications. The significance of this report lies in its contribution to the differential diagnosis of similar cases. Future studies involving larger cohorts with extended follow-up periods are necessary to validate these findings and establish comprehensive management guidelines.

Although rare, solitary neurofibromas should be considered in the differential diagnosis of atypical toe tip ulcers, particularly in patients with diabetes. This case highlights the critical need for vigilance and thorough evaluation in the management of foot lesions to avoid misdiagnosis and ensure appropriate treatment. Despite the rarity of such cases, this report underscores the importance of considering neurofibromas in differential diagnoses and provides valuable insights into their management. Further research should aim to elucidate the characteristics and optimal management strategies for neurofibromas in atypical locations to ensure effective diagnosis and treatment in future cases.

## Figures and Tables

**Figure 1 medicina-60-01200-f001:**
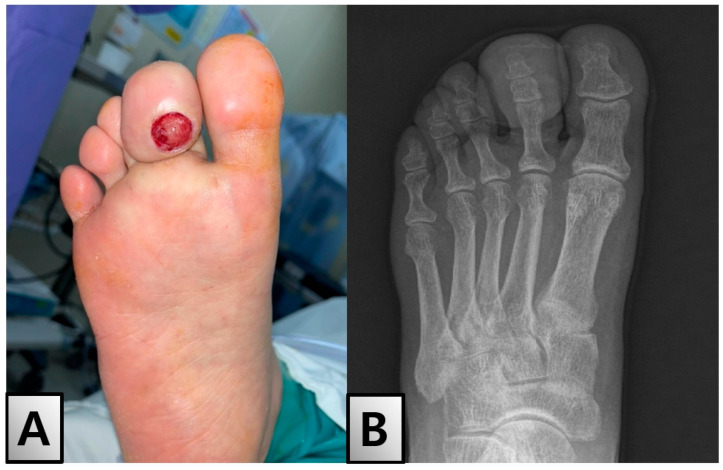
Initial Presentation and Diagnostic Imaging. (**A**) Clinical photograph of the patient’s right second toe showing an ulcerated lesion on the tip with significant induration and swelling. The ulcer is round, well-circumscribed, and exhibits a red granular base. (**B**) X-ray image of the right foot demonstrating no specific bony abnormalities associated with the ulcerated lesion. The bony structures of the foot, including the phalanges and metatarsals, appear normal without signs of osteomyelitis or fracture.

**Figure 2 medicina-60-01200-f002:**
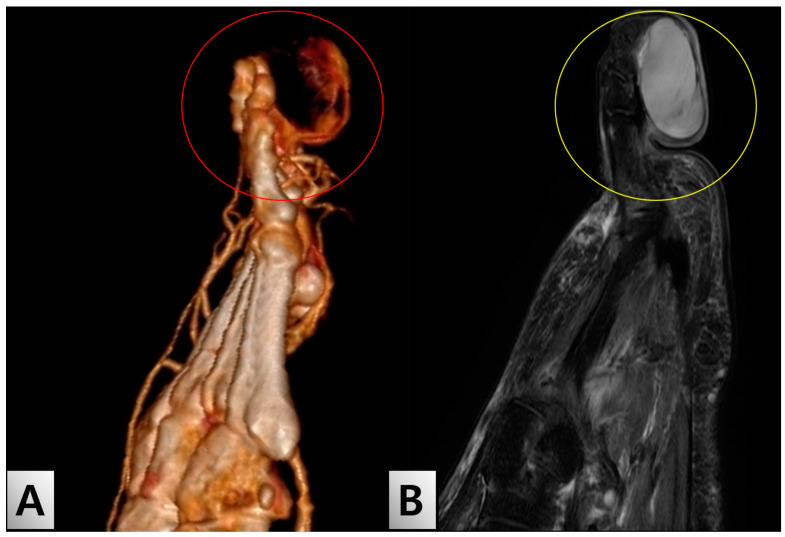
Diagnostic Imaging of the Right Second Toe. (**A**) Lower extremity angio CT showing a mass-like lesion with feeding vessels (red circle), indicating significant vascular involvement in the lesion. (**B**) MRI finding demonstrating a cystic lesion (yellow circle) at the tip of the right second toe. The MRI shows the extent of the lesion, suggesting a soft tissue tumor rather than a typical diabetic ulcer. The well-demarcated margins and internal characteristics of the lesion are consistent with a neurofibroma. Abbreviations: CT, computed tomography; MRI, magnetic resonance imaging.

**Figure 3 medicina-60-01200-f003:**
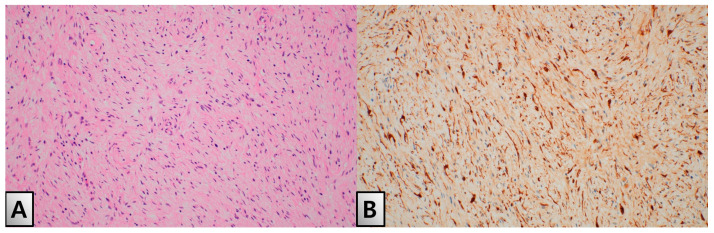
Histopathological examination confirmed the presence of neurofibroma, characterized by spindle cells within a myxoid stroma, and showed S-100 protein positivity. (**A**) Hematoxylin and eosin (H&E) staining at 100x magnification, illustrating the spindle cells within a myxoid stroma. (**B**) Immunohistochemical staining for S-100 protein at 100x magnification, demonstrating strong positivity in the spindle cells.

**Figure 4 medicina-60-01200-f004:**
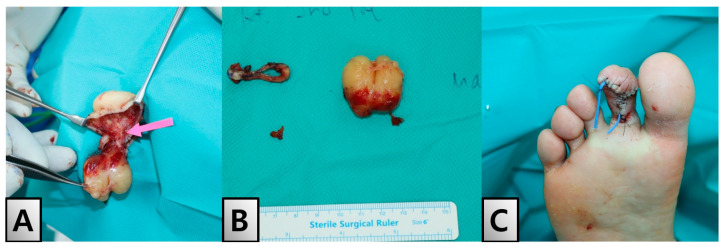
Intraoperative Findings and Immediate Postoperative Outcome. (**A**) Intraoperative finding showing the mass encapsulated by a feeding vessel, which is being meticulously dissected and identified (pink arrow) on the right second toe. (**B**) The excised mass, displaying a sheath encasing the neurofibroma. (**C**) Immediate postoperative view of the right second toe demonstrating successful reconstruction using Z-plasty, effectively replicating the original appearance of the toe.

**Figure 5 medicina-60-01200-f005:**
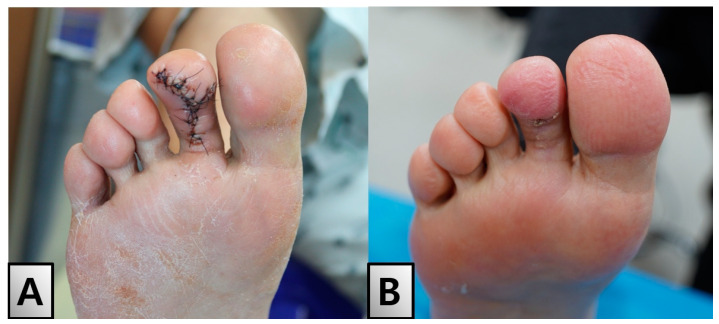
Postoperative Healing and Follow-up. (**A**) Photograph taken on postoperative day 5 showing a clean surgical site on the right second toe. The patient was discharged without any postoperative complications. (**B**) Follow-up photograph taken two months postoperatively demonstrating successful healing without recurrence or complications. The toe appears well healed with no signs of infection or other adverse effects.

**Table 1 medicina-60-01200-t001:** Laboratory Test Results.

Parameter	Result	Reference Range
HbA1c (%)	6.4	4.0–5.6
Fasting Blood Glucose (mg/dL)	180	70–100
CRP (mg/L)	12.5	0–10
WBC (×10^9^/L)	8.6	4.0–11.0
Total Cholesterol (mg/dL)	190	<200
LDL Cholesterol (mg/dL)	99	<100
HDL Cholesterol (mg/dL)	41	>40
Triglycerides (mg/dL)	145	<150
Creatinine (mg/dL)	0.9	0.6–1.2
Blood Urea Nitrogen (BUN) (mg/dL)	15	7-20
Albumin (g/dL)	4	3.5–5.0
Alanine Aminotransferase (ALT) (IU/L)	25	7–56
Aspartate Aminotransferase (AST) (IU/L)	22	10–40
Hemoglobin (g/dL)	13.5	13.8–17.2 (males)
		12.1–15.1 (females)
Platelets (×10^9^/L)	250	150–450

Abbreviations: HbA1c, Hemoglobin A1c; CRP, C-Reactive Protein; WBC, White Blood Cell Count; LDL, Low-Density Lipoprotein; HDL, High-Density Lipoprotein.

## Data Availability

The data presented in this study are available on reasonable request from the corresponding author.

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
