# Peer review of "A Rare Case of Solitary Neurofibroma Misdiagnosed as Diabetic Foot Ulcer in the Toe Tip Region"

_medicina, 2024, doi:10.3390/medicina60081200_

Round 1
Reviewer 1 Report
Comments and Suggestions for Authors
This paper presents a case report of a neurofibroma in the second pododactyl of a diabetic patient, without vascular or peripheral neurological involvement.
The case was very well managed with imaging, laboratory and biopsy studies. It presents a result that is not very common, with an interesting outcome after the surgical approach. It is well-written and has a solid literature base. I suggest publishing it as presented.
Author Response
Comment: This paper presents a case report of a neurofibroma in the second pododactyl of a diabetic patient, without vascular or peripheral neurological involvement. The case was very well managed with imaging, laboratory, and biopsy studies. It presents a result that is not very common, with an interesting outcome after the surgical approach. It is well-written and has a solid literature base. I suggest publishing it as presented.
Response: We are delighted that reviewer found our manuscript well-written, well-managed, and based on solid literature. We sincerely appreciate the positive feedback and thank the reviewer for their insightful comments.
Reviewer 2 Report
Comments and Suggestions for Authors
I am grateful for the opportunity to review the article "A Rare Case of Solitary Neurofibroma Misdiagnosed as Dia- 2 betic Foot Ulcer in the Toe Tip Region".
The aforementioned case of a 55-year-old woman with type 2 diabetes and an initially incorrectly diagnosed diabetic foot ulcer is highly intriguing and worth the reader's attention.
I would like to focus authors' attention on several factors that, when considered, will considerably enhance the quality of the manuscript:
1. in lines 50-51, the authors write about type 2 diabetes and its duration, but there is no information about the method of its treatment (diet? non-insulin pharmacotherapy? insulin?), which information has a significant impact on the patient's clinical condition - please to add appropriate information.
2. in lines 69-71, the authors write about normal laboratory test results, except for slightly increased inflammation markers - I would suggest providing specific data, e.g. in tabular form, with particular emphasis on ilaboratory parameters outside the reference values ​​and detailing data on metabolic control of diabetes (including HbA1c percentage and glycemia upon admission to hospital).
3. the case report lacks information about antidiabetic pharmacotherapy in the perioperative period (what partly refers to point 1), and whether the procedure had an impact on the metabolic control of diabetes - do the authors have such data? if so, please include them in the case descriptions.
The discussion is conducted in a concise but correct manner. The authors' consideration of the manuscript's limitations is commendable.
The citations presented by the authors are unquestionable.
I strongly advise the authors to consider the comments previously mentioned. I would be delighted to review the authors' response and the revised manuscript.
I congratulate the authors of the described case and wish them further success in their scientific path.
Author Response
Comment 1:
In lines 50-51, the authors write about type 2 diabetes and its duration, but there is no information about the method of its treatment (diet? non-insulin pharmacotherapy? insulin?), which information has a significant impact on the patient's clinical condition - please add appropriate information.
Response 1:
We have added information regarding the patient’s diabetes treatment method. The patient was on non-insulin pharmacotherapy, specifically metformin, and received dietary education.
Revised Text:
"A 55-year-old woman with a history of type 2 diabetes mellitus, diagnosed 5 months prior and managed with metformin and dietary education, presented with a gradually enlarging ulcer on the right second toe tip over the past 2 months."
Comment 2:
In lines 69-71, the authors write about normal laboratory test results, except for slightly increased inflammation markers - I would suggest providing specific data, e.g., in tabular form, with particular emphasis on laboratory parameters outside the reference values and detailing data on metabolic control of diabetes (including HbA1c percentage and glycemia upon admission to hospital).
Response 2:
We have included a table with specific laboratory test results, highlighting any parameters outside the reference values and detailing data on the metabolic control of diabetes, including HbA1c percentage and glycemia upon admission.
Revised Text and Table:
" Laboratory tests showed that blood parameters were within normal limits, except for elevated levels of HbA1c and fasting blood glucose, as well as slightly increased levels of inflammatory markers. Detailed laboratory results, including Hemoglobin A1c(HbA1c) and glycemia upon admission, are presented in Table 1."
Table 1. Laboratory Test Results
|
Parameter |
Result |
Reference Range |
|
HbA1c (%) |
6.4 |
4.0 - 5.6 |
|
Fasting Blood Glucose (mg/dL) |
180 |
70 - 100 |
|
CRP (mg/L) |
12.5 |
0 - 10 |
|
WBC (x10^9/L) |
8.6 |
4.0 - 11.0 |
|
Total Cholesterol (mg/dL) |
190 |
<200 |
|
LDL Cholesterol (mg/dL) |
99 |
<100 |
|
HDL Cholesterol (mg/dL) |
41 |
>40 |
|
Triglycerides (mg/dL) |
145 |
<150 |
|
Creatinine (mg/dL) |
0.9 |
0.6 - 1.2 |
|
Blood Urea Nitrogen (BUN) (mg/dL) |
15 |
7-20 |
|
Albumin (g/dL) |
4 |
3.5 - 5.0 |
|
Alanine Aminotransferase (ALT) (IU/L) |
25 |
7 - 56 |
|
Aspartate Aminotransferase (AST) (IU/L) |
22 |
10 - 40 |
|
Hemoglobin (g/dL) |
13.5 |
13.8 - 17.2 (males) |
|
Platelets (x10^9/L) |
250 |
150 - 450 |
Abbreviations: HbA1c; Hemoglobin A1c, CRP; C-Reactive Protein, WBC; White Blood Cell Count, LDL; Low-Density Lipoprotein, HDL; High-Density Lipoprotein
Comment 3:
The case report lacks information about antidiabetic pharmacotherapy in the perioperative period (what partly refers to point 1), and whether the procedure had an impact on the metabolic control of diabetes - do the authors have such data? If so, please include them in the case descriptions.
Response 3:
We have included information regarding the antidiabetic pharmacotherapy in the perioperative period and the impact of the procedure on the metabolic control of diabetes.
Revised Text:
"The patient continued her metformin regimen and dietary education throughout the perioperative period, except during the use of contrast agents for CT imaging. There were no significant fluctuations in blood glucose levels during her hospital stay, and her HbA1c remained stable postoperatively."
Round 2
Reviewer 2 Report
Comments and Suggestions for Authors
Thank you for the opportunity to again review the article "A Rare Case of Solitary Neurofibroma Misdiagnosed as Diabetic Foot Ulcer in the Toe Tip Region".
The authors of the article responded to the reviewer's comments and introduced changes that, in my opinion, made the manuscript more readable and reader-friendly.
I suggest accepting the article for publication in its current format.